# Existing gaps and missed opportunities in delivering quality nutrition services in primary healthcare: a descriptive analysis of patient experience and provider competence in 11 low-income and middle-income countries

Marwa Ramadan ![ORCID], Tonny B Muthee, Latifat Okara ![ORCID], Cameron Feil ![ORCID], Manuela Villar Uribe ![ORCID]

Health, Nutrition and Population, The World Bank Group, Washington, District of Columbia, USA

**Correspondence to**
Dr Marwa Ramadan;
marwa.eldesoky1987@gmail.com

## ABSTRACT

**Objectives** To assess the competence of primary healthcare (PHC) providers in delivering maternal and child nutrition services at the PHC level and patients' experience in receiving the recommended components of care.

**Design** Observational cross-sectional analysis.

**Setting** Healthcare facilities in low/middle-income countries (LMICs) with available service provision assessment surveys (Afghanistan (2018), Democratic Republic of Congo (2018), Haiti (2017), Kenya (2010), Malawi (2013–2014), Namibia (2009), Nepal (2015), Rwanda (2007), Senegal (2018), Tanzania (2015) and Uganda (2007).

**Participants** 18 644 antenatal visits and 23 262 sick child visits in 8458 facilities across 130 subnational areas in 11 LMICs from 2007 to 2019.

**Outcomes** (1) Provider competence assessed as the direct observations of actions performed during antenatal care (ANC) and sick child visits; and (2) patients' experience defined as the self-reported awareness of the nutrition services received during ANC and sick child visits and provider effectiveness in delivering these services.

**Results** Except for DRC, all countries scored below 50% on patients' experience and provider competence. More than 70% of clients were advised on taking iron supplements during pregnancy; however, less than 32% of patients were advised on iron side effects in all the studied countries. Across all countries, providers commonly took anthropometric measurements of expectant mothers and children; however, such assessments were rarely followed up with advice or counselling about growth patterns. In addition, less than 20% of observed providers advised on early/immediate breast feeding in all countries with available data.

**Conclusion** The 11 assessed countries demonstrated the delivery of limited nutrition services; nonetheless, the apparent deficiency in the extent and depth of questions asked for the majority of tracer activities revealed significant opportunities for improving the quality of nutrition service delivery at the PHC level.

## STRENGTHS AND LIMITATIONS OF THIS STUDY

⇒ This study uses standardised publicly available health facility surveys to capture both the patient and the provider perspectives in delivering quality nutrition services in 11 low-income and middle-income countries.

⇒ Rather than solely focusing on the availability of nutrition services through system inputs, this study assesses the service delivery processes that underpin the quality of nutrition services at the primary healthcare (PHC) level.

⇒ This study only examined the quality of nutrition services in antenatal care and child health visits, data and tools were inadequate for other key PHC services as non-communicable diseases.

⇒ This study is descriptive in nature; inferential statistics were not used to statistically test or account for potential temporal and geographic variations across countries.

⇒ The tools used in the present analysis only focused on facility-based nutrition services with inability to capture community-based nutrition services.

## BACKGROUND

Over the past 20 years, significant progress has been made in nutrition-related outcomes in many low/middle-income countries (LMICs). Many LMICs, however, now face the dual malnutrition burden of undernutrition and overweight, posing significant challenges for progress in nutrition, population health and human capital development. Globally, it is estimated that one out of every nine people is hungry or undernourished and that one in every three people is overweight or obese.[1][2] Furthermore, the food insecurity caused by the COVID-19 pandemic and ongoing conflicts such as in Ukraine

and Russia are additional threats to nutrition-related progress. The United Nations estimates that 2.37 billion people lacked access to adequate food sources in 2020, an increase of 320 million people compared with 2019. The combined effects of the pandemic, and conflict-related turmoil if left unmitigated, will likely result in longstanding and severe consequences for health and nutrition.[3 4]

Primary healthcare (PHC) systems are uniquely poised to address the problems of undernutrition and malnutrition in LMICs, especially as the world continues to face the challenges of COVID-19. PHC acts as the first point of contact between patients and the health system, allowing healthcare providers to identify health risks early and to deliver personalised care over a person's lifetime. PHC is delivered at the community level, therefore in a position to identify and empanel patients, provide outreach services and continuously monitor nutrition-related indicators.[5–7] Integrating essential nutrition services into routine PHC services, specifically in maternal, reproductive, newborn and child health services (RMNCH), has the potential to greatly impact population health as good nutrition is a determinant of improved health outcomes.[8]

The majority of PHC and nutrition-related research has focused on the availability of health system inputs (eg, availability of drugs, equipment, workforce and information systems at the facility). However, the extent to which nutrition services are integrated into routine PHC in LMICs, along with the quality of these services, is not well known. Specifically, there is a significant gap in understanding patients' experience in receiving the recommended components of high-quality care and whether nutrition services in LMICs are delivered by competent providers.[9 10]

Health facility surveys such as the Service Delivery Indicators (SDI), Service Availability and Readiness Assessment (SARA) and the Service Provision Assessment (SPA) do collect information pertaining to the quality of nutrition services provided in many LMICs.[11] The latter can be used to better understand key aspects of the service delivery process from both the patients' and providers' sides. This is evidenced by King *et al*,[11] who proposed a standardised set of measures/indices for nutrition service quality among pregnant women and children based on expert review and data available in standardised health facility surveys such as SARA and SPA.

Given that the focus of the existing literature has been on the availability of nutrition services through system inputs,[9 12 13] this study bridges the gap by focusing on the service delivery processes that underpin the quality of nutrition services at the PHC level. Precisely, we assess the competence of PHC providers in delivering essential and preventive nutrition services during antenatal care (ANC) and sick child visits and patients' experience in receiving the recommended components of care.

## METHODS

### Data sources

In this observational cross-sectional study, we used the SPA survey for information on nutrition services provided during PHC visits. The SPA survey is a health facility assessment that provides a comprehensive overview of a country's health service delivery.[14] It provides insight into the availability of various health services, service readiness to provide these services, quality of care (the extent to which the service delivery process follows generally accepted standards of care) and client satisfaction with the services offered.[14] A total of 12 countries completed a SPA survey from 2007 to 2019, 11 of which had sufficient data for the present analysis. The final analytic dataset included anonymised data from the following SPA surveys: Afghanistan (2018), Democratic Republic of Congo (2018), Haiti (2017), Kenya (2010), Malawi (2013–2014), Namibia (2009), Nepal (2015), Rwanda (2007), Senegal (2018), Tanzania (2015) and Uganda (2007). Bangladesh was excluded due to the limited availability of information on several tracer items necessary for the analysis.

### Metrics

In this study, we defined PHC as services provided in outpatient consultations in public and private non-hospital SPA facilities. We specifically focused on two critical processes for high-quality service delivery: patients' experience and provider competence. For each critical process, tracer items were identified based on the set quantitative measures/tracer items of nutrition service quality for pregnant women and children proposed by King *et al*,[11] the list of SPA tracer items for measuring facility service delivery of nutrition services,[12] as well as SPA data availability across the studied countries (see table 1). Patients' experience was assessed as the self-reported awareness of the nutrition services received in PHC and provider effectiveness in delivering these services. As such, patient education and support through counselling were a key focus. The client exit interview, a SPA tool in which clients provide their perceptions on the visit, was used to collect information on the patient's experience of the services offered. Twelve tracer items for patient experience were extracted from the SPA datasets across two service areas: ANC and sick child consultations. Similarly, provider competence was assessed as the direct observations of actions performed during ANC and sick child visits. Seventeen tracer activities for provider competence were extracted from SPA datasets. The observation checklist was the main SPA tool used to collect information on the identified tracer activities. Table 1 provides a breakdown of SPA tracer activities/indicators and SPA tools used in the present analysis. Detailed information on each of the used SPA tools are provided by the demographic and health surveys programme and publicly available on its website.[15]

**Table 1** List of patients' experience and provider competence tracer activities

| Quality subdomain | Thematic area | Tracer activities | SPA tool |
|---|---|---|---|
| Patients' experience | Maternal | 1. Client knows at least one side effect of iron<br>2. Provider encouraged questions<br>3. Provider advised 6 months exclusive breast feeding<br>4. Provider ever advised breast feeding<br>5. Provider explained iron side effects in any ANC visit<br>6. Provider explained how to take iron/folate in any ANC visit<br>7. Clients reported being advised on nutrition during pregnancy | Client exit interview—Antenatal care |
| | Child | 1. Provider asked about child's usual eating habits<br>2. Child was weighed at the facility (exit)<br>3. Provider counselled on appropriate fluid/breast feeding<br>4. Provider counselled on appropriate solid food intake<br>5. Provider discussed child's weight and growth | Client exit interview—Curative childcare and growth monitoring services |
| Provider competence | Maternal | 1. Provider advised exclusive breast feeding<br>2. Provider advised early/immediate breast feeding<br>3. Provider described how to take iron pills<br>4. Provider mentioned side effects of iron<br>5. Provider explained purpose of iron pills<br>6. Provider discussed nutrition during pregnancy<br>7. Provider checked pallor of conjunctiva/palms for pallor<br>8. Provider measured weight during ANC | Observation checklist—Antenatal care |
| | Child | 1. Provider counselled on child weight/growth chart<br>2. Provider weighed child<br>3. Provider plotted weight on growth chart<br>4. Provider checked palms/conjunctiva/mouth for pallor<br>5. Provider asked/caretaker mentioned if child unable to drink or breast feed<br>6. Provider asked about normal (breast)feeding pattern<br>7. Provider asked about (breast)feeding pattern during this illness<br>8. Provider asked if child received any deworming<br>9. Provider asked if child received vitamin A within 6 months | Observation checklist—Curative childcare and growth monitoring services |

ANC, antenatal care; SPA, Service Provision Assessment.

## Data analysis

To assess patients' experience and competence of providers in delivering nutrition services during PHC visits, each of the extracted relevant tracer items (table 1) was recoded to a binary (0, 1) variable/indicator, where a value of 1 indicated that the tracer item was available or provided, and a value of 0 indicates that the tracer item was not available or provided. Next, we calculated the facility-level averages for each of the selected tracer items. Given that SPA surveys are hierarchical in nature where country data is collected at three levels (the facility, the provider and the client), we applied the country-specific survey weights for all levels to ensure that national averages were representative per country. The only exception was in Afghanistan, where SPA data collection was limited to the urban provinces, so national averages were not representative of the entire primary healthcare facilities in the country. We reported national estimates for individual tracer items per country. In addition, we calculated two summary scores for each of patient experience and provider competence subdomains. Summary scores were calculated as simple unweighted averages of the 12 tracer items for patients' experience and the 17 tracer items for providers' competence. In Namibia and Rwanda, providers' competence subdomain score was based on 15 tracer items instead of 17 as data were not available for the two tracer items: (1) provider's advice on early/immediate breast feeding and (2) provider asked if the child received any deworming medications. In Uganda, provider competence summary score was based on 14 items only as three tracer items were missing: (1) provider's advice on early/immediate breast feeding, (2) provider asked if the child received any deworming medications and (3) the provider checked palms/conjunctiva/mouth for pallor. Meanwhile, all countries had 12 tracer items for patient experience except Namibia where clients' reports on the assessment of child eating habits by a provider were missing. Stata V.17 and R were used to analyse data and generate figures.

## Patient and public involvement

This study relied on secondary analysis of publicly available datasets. Therefore, patients and the public were not involved in the formulation of this research.

**Table 2** Descriptive characteristics of the analysed sample

| | Primary* (%) | Public (%) | Rural (%) | Facilities (n) | Districts (n) | Regions (n) |
|---|---|---|---|---|---|---|
| Afghanistan | 38 | 23 | 1 | 160 | – | 7 |
| DRC | 54 | 61 | 76 | 1412 | 502 | 26 |
| Haiti | 87 | 34 | 62 | 1033 | 34 | 10 |
| Kenya | 64 | 50 | – | 695 | 142 | 8 |
| Malawi | 89 | 48 | 68 | 1060 | 28 | 3 |
| Namibia | 89 | 74 | – | 411 | 4 | 13 |
| Nepal | 73 | 78 | – | 992 | 75 | 5 |
| Rwanda | 92 | 57 | – | 538 | – | 5 |
| Senegal | 92 | 81 | 21 | 466 | – | 14 |
| Tanzania | 78 | 67 | 63 | 1200 | 10 | 30 |
| Uganda | 76 | 71 | – | 491 | – | 9 |
| Total | 76 | 59 | 39 | 8458 | 1297 | 130 |

Shaded vaues signifies that no data are available.
*Primary level was defined as services provided in outpatient consultations in public and private non-hospital Service Provision Assessment facilities.

## RESULTS

The present study included information on the quality of nutrition services in 18 644 ANC visits and 23 262 sick child visits in 8458 facilities across 130 subnational areas in 11 LMICs. More than half of the analysed facilities (59%, n=4990) were public, 76% (n=6248) were classified as primary/non-hospital, and 39% (n=3298) of the facilities with available data were located in rural areas (table 2). For this study, analysis was limited to primary (non-hospital facilities). All the studied countries, except DRC, scored below 50% in both patients' experience and provider competence (figure 1). For example, based on the 12 tracer activities assessed for patient experience, 36% of the interviewed clients (on average) reported being aware of the nutrition services received in PHC and confirmed provider effectiveness in delivering these services. Similarly, only about 31% of the observed providers (on average) successfully and effectively delivered essential nutrition services in PHC based on the 17 tracer activities assessed for provider competence. Generally, patient experience summary scores were higher than provider competence scores in 7 out of the 11 studied countries. Exceptions were observed in Kenya, Namibia, Rwanda and Uganda. However, in three out of the four countries (Namibia, Rwanda and Uganda), provider competence score was calculated using fewer tracer items as indicated in the methods section. Furthermore, all four countries conducted the SPA survey prior to 2013 so the scores may not be comparable to recent surveys.

Assessment of individual tracer items of patients' experience with maternal nutrition services revealed that more than half of clients were encouraged to ask nutrition questions during ANC visits in 9 out of the 11 studied countries. In addition, more than 70% of clients in each of the studied countries reported being advised on taking iron supplements during pregnancy. However, less than one-third (32%) of patients reported being advised on iron side effects in each of the studies countries. In contrast, when asked in an exit interview, more than 45% of clients knew at least one side effect of iron in 7 out of the 11 countries assessed. In addition, less than 34% of clients reported being advised on exclusive breast feeding in each of the studied countries, and only in Malawi and Namibia, more than 50% of clients were ever advised on breast feeding during ANC (table 3).

Patients' experience with child nutrition services was found to be significantly lagging compared with maternal services. In 5 out of the 11 countries analysed, more than 69% of clients reported weighing the child at the facility; however, in all the studied countries, less than 42% of clients reported that the child's weight and growth were discussed. Moreover, in 8 out of the 10 countries with available data, less than 25% of clients reported being asked about the child's eating habits. In addition, in the majority of studied countries (10 out of 11), at most, 40% of clients were counselled on appropriate fluid intake or breast feeding, and less than 40% were counselled on appropriate solid food intake. The only exception was DRC, where almost all clients (99%) reported being advised on both appropriate fluid and appropriate solid food intake (table 3).

Assessment of tracer activities for providers' competence in maternal nutrition services revealed that weight was measured during the ANC consultation for the majority (more than 70%) of clients in 10 out of the 11 analysed countries. The only exception was in Afghanistan, where only 26% of observed providers measured the mother's weight. Signs of anaemia were also checked by more than half of providers in 8 out of the 10 countries with available data. However, in 7 out of the 11 countries analysed, less than half of observed providers explained the purpose of iron or how to take iron pills. Furthermore,

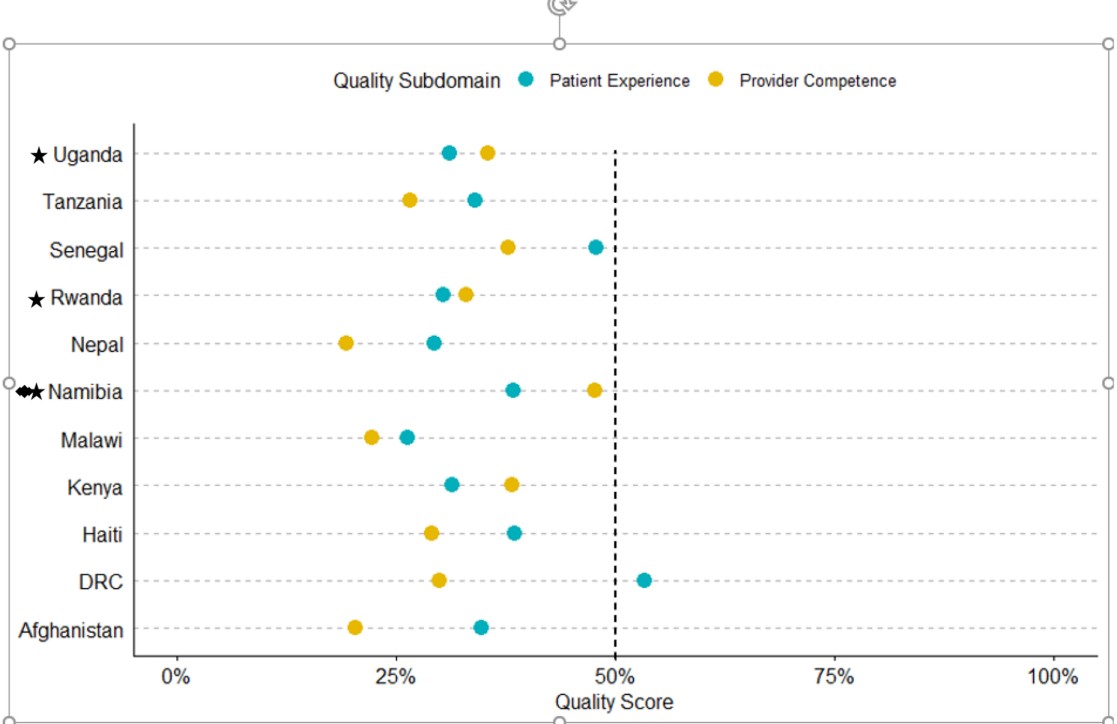

★Provider competence summary score is calculated as the average of 17 tracer items extracted from the Service Provision Assessment (SPA) survey in all 11 countries except Namibia, Rwanda, and Uganda. In Namibia and Rwanda, the provider competence subdomain was missing 2 tracer items (the provider advised early/immediate breastfeeding , the provider asked if the child received a deworming medication) and summary score was calculated as the average of the 15 remaining tracer items, and in Uganda, provider competence subdomain was missing 3 tracer items (the provider advised early/immediate breastfeeding , the provider asked if the child received a deworming medication, the provider checked for pallor for conjunctiva and/or pallor for palms)and the summary score was calculated as the average of the remaining 14 tracer items. ☛ Patient experience was calculated as the average of 12 tracer items extracted from the Service Provision Assessment (SPA) survey in all 11 countries except Namibia where clients' reports on the assessment of child eating habits by a provider were missing

**Figure 1** Quality of nutrition services by subdomain score in each of the studied countries.

less than 15% of providers mentioned the side effects of iron in all the studied countries. Counselling on breast feeding was also one of the areas with significant gaps. Specifically, less than 20% of observed providers advised on early/immediate breast feeding in all countries with available data (eight countries), and less than 35% of observed providers advised exclusive breast feeding in all the 11 studied countries (table 4).

Similarly, measuring weight or checking for signs of anaemia during sick child consultations were among the highest-scored tracer activities of provider competence. However, less than one-third (30%) of observed providers counselled clients on child growth in each of the studied countries and less than one quarter (21%) plotted the child weight on a growth chart in 9 out of the 11 countries studied. In addition, less than 45% of observed providers in each of the analysed countries asked about the child's normal feeding pattern or feeding pattern during illness. Furthermore, in 8 out of the 11 countries analysed, less than 30% of observed providers asked whether the child was unable to feed or drink during sick child

consultations. Also, whenever data was available, 32% of providers, at most, asked if the child received vitamin A or a deworming medication in each of the studied countries (table 4).

## DISCUSSION

Overall, the descriptive analysis of patients' experience and provider competence in the present study showed the delivery of low-quality maternal and child nutrition services at the PHC level across 11 LMICs, where only one country (DRC) reported a provider competence score above 50%. These findings highlight the urgent need to integrate quality essential and preventive nutrition services into routine PHC in LMICs.

We also found compelling evidence indicating the lack of depth in providers' assessment of the nutritional status of expectant mothers and children. Across all assessed countries, providers commonly took anthropometric measurements of expectant mothers and children; however, such assessments were not always followed up

**Table 3** Tracer items of patient experience with maternal and child nutrition services

| | Patients' experience tracer items | AF | DRC* | HT | KE* | MW | NM* | NP | RW* | SN | TZ | UG* |
|---|---|---|---|---|---|---|---|---|---|---|---|---|
| Maternal | Client knows at least one side effect of iron | 70.1 (13.5) | 48.8 (3.7) | 74.8 (4.1) | 45 (3.5) | 73.6 (3.6) | 25.4 (2.4) | 18.7 (2.1) | 7.8 (1.7) | 80.1 (3.8) | 59.5 (2.5) | 21.4 (3.0) |
| | Provider explained iron side effects in any ANC visit | 30.9[9] | 20.8 (1.6) | 16.3 (1.7) | 18.1 (2.8) | 16.4 (1.4) | 29.3 (2.6) | 21.8 (2.2) | 11.6 (2.4) | 32.4 (2.9) | 24.8 (1.4) | 17.2 (3.1) |
| | Provider explained how to take iron/folate in any ANC visit | 72.1 (8.4) | 88 (1.3) | 86 (1.5) | 94.5 (1.7) | 96.2 (0.6) | 95 (1.2) | 95.5[1] | 86.8 (2.9) | 91.9 (1.5) | 96.1 (0.6) | 87.5 (0) |
| | Provider encouraged questions | 23.9 (9.3) | 53.5[2] | 59.7 (2.5) | 67.5 (3.1) | 79.3 (1.9) | 64.4 (3.1) | 35.7 (2.5) | 56.7 (3.5) | 69.9 (3.1) | 77.6 (1.5) | 59.9 (3.7) |
| | Provider advised 6 months exclusive breast feeding | 1.1 (1.5) | 20.4 (1.4) | 16 (1.5) | 28.8 (2.6) | 34 (1.7) | 8.4 (1.5) | 12.9 (1.7) | 28.2 (2.7) | 11.8 (1.8) | 25.2 (1.4) | 16.3 (2.7) |
| | Provider ever advised breast feeding | 3.6 (2.8) | 28.1 (1.6) | 27.6 (1.9) | 33.1 (2.6) | 52.7 (1.7) | 61.1 (2.9) | 21.2 (2.1) | 32.3 (2.8) | 14.5 (1.9) | 34.6 (1.5) | 28.7 (3.4) |
| | Provider advised on nutrition during pregnancy | 63.3 (8.2) | 37.9 (1.7) | 45.9 (2.2) | 51.8[3] | 63.3 (1.9) | 60 (2.9) | 77.3 (2.2) | 37.8 (2.8) | 52.7 (3.3) | 43.4 (1.6) | 50.1 (3.7) |
| Child | Provider discussed child's weight and growth | 39.5 (6.2) | 33 (1.8) | 34.2 (1.6) | 22.8 (1.9) | 16.1[1] | 33.5 (1.9) | 16.2 (1.4) | 42.3 (2.1) | 37.6 (2.8) | 37.8 (1.3) | 20 (2.2) |
| | Provider counselled on appropriate solid food intake | 38.6 (5.8) | 99.6 (0.2) | 15.4 (1.2) | 16.5 (1.7) | 8.8 (0.7) | 21.7 (1.6) | 18.2 (1.5) | 10.8 (1.1) | 37.7 (2.8) | 12.2 (0.8) | 24.6 (2.4) |
| | Provider counselled on appropriate fluid/breast feeding | 38.9 (5.4) | 99.7 (0.2) | 25 (1.4) | 19.7 (1.7) | 12.4 (0.8) | 25.6 (1.6) | 26.6 (1.6) | 12.3 (1.2) | 30.9 (2.7) | 19.1[1] | 40 (2.8) |
| | Provider asked about child's usual eating habits | 32 (5.2) | 22 (1.5) | 25.3 (1.4) | 24.8[2] | 11.8 (0.8) | – | 12.5 (1.2) | 9.9 (1.1) | 37.9 (2.8) | 22.0 (1.0) | 18.2 (2.1) |
| | Child was weighed at the facility (exit) | 49.5 (7.8) | 75.9 (1.7) | 86.9 (1.3) | 44.7 (2.5) | 29.2 (1.5) | 88.4 (1.6) | 54.0 (2.0) | 69.4 (1.9) | 94.6 (1.3) | 31 (1.3) | 46.5 (3.0) |

SEs are in parentheses. AF: Afghanistan 2018, DRC: Democratic Republic of Congo 2018, HT: Haiti 2017, KE: Kenya 2010, MW: Malawi 2013, NM: Namibia 2009, NP: Nepal 2015, RW: Rwanda 2007, SN: Senegal 2017, TZ: Tanzania 2015, UG: Uganda 2007.

Shaded vaues signifies that no data are available.

*Countries with older Service Provision Assessment surveys (prior to 2013 so data may not be comparable). Analysis is limited to primary (non-hospital) health facilities. Maternal experience was only assessed in facilities offering antenatal care services. Child experience was assessed in facilities offering services for under-five children.

ANC, antenatal care.

**Table 4** Providers' competence in maternal and child nutrition services

| Provider competence tracer items | AF | DRC* | HT | KE* | MW | NM* | NP | RW* | SN | TZ | UG* |
|---|---|---|---|---|---|---|---|---|---|---|---|
| **Maternal** | | | | | | | | | | | |
| Provider checked pallor of conjunctiva/palms for pallor | 28.4 (8.3) | 56 (1.8) | 51.8 (2.4) | 79.9 (2.7) | 78.1 (1.9) | 58.9 (3.3) | 36.8 (2.5) | 78.5 (2.9) | 88.1 (1.9) | 54.5 (1.8) | – |
| Provider described how to take iron pills | 49.5 (9.7) | 37.5 (1.8) | 37 (2.3) | 42.6 (3.3) | 58.6 (2.8) | 65.8 (2.8) | 24.3 (2.1) | 35.6 (3.3) | 50.2 (3.1) | 58.7 (1.7) | 48.8 (3.7) |
| Provider explained purpose of iron pills | 47.8 (9.4) | 29 (1.7) | 30.2 (2.2) | 39 (3.2) | 57.4 (2.3) | 53.5 (3.1) | 21.6 (2.2) | 31.6 (3.2) | 61 (3.1) | 59.7 (1.7) | 31.4 (3.6) |
| Provider mentioned side effects of iron | 16.2 (8.4) | 4.3 (0.7) | 3.4 (0.8) | 11.7 (2.2) | 7.3 (1.1) | 12.0 (2.0) | 4.4 (1.0) | 13.3 (2.3) | 9.3 (1.7) | 9.7 (1.0) | 5.7 (1.9) |
| Provider advised exclusive breast feeding | 0 (0) | 10.8 (1.2) | 5.3 (1.1) | 23.4 (2.7) | 5.5 (1.1) | 34.4 (3.2) | 1.5 (0.7) | 20.5[3] | 4.2 (1.2) | 11.6 (1.1) | 15.9 (2.8) |
| Provider advised early/immediate breast feeding | 1.8 (3.0) | 7.2 (1.0) | 2.2 (0.7) | 18.1 (2.5) | 3.2 (0.8) | – | 2.7 (0.9) | – | 5.2 (1.2) | 7.0 (0.8) | – |
| Provider discussed nutrition during pregnancy | 58.4[9] | 28 (1.7) | 50.2 (2.3) | 45.3 (3.2) | 35.3 (2.3) | 46.3 (3.2) | 55.9 (2.5) | 40.3 (3.4) | 54.6 (3.1) | 31.4 (1.5) | 32.6 (3.5) |
| Provider measured weight during ANC | 25.4 (8.9) | 74.5 (1.8) | 86.1 (1.8) | 92.6 (1.8) | 75.5 (2.2) | 93.5 (1.8) | 76.2 (2.3) | 98.9 (0.6) | 92.7 (1.3) | 79.6 (1.5) | 73.8 (3.9) |
| **Child** | | | | | | | | | | | |
| Provider weighed child | 47.5 (7.4) | 72.7 (1.8) | 83 (1.8) | 42.2 (2.5) | 17.6 (1.3) | 88.1 (1.7) | 45.5 (2.0) | 63.4 (2.2) | 94.3 (1.3) | 17.1 (1.1) | 45.3 (3.0) |
| Provider plotted weight on growth chart | 3.4 (1.7) | 5.5 (0.9) | 9.7 (1.1) | 21.2 (1.8) | 3.9 (0.6) | 48.6 (2.4) | 19.4 (1.6) | 8.5 (1.1) | 54.9 (3.2) | 4.9 (0.6) | 12.9[2] |
| Provider counselled on child weight/growth chart | 13.8 (4.9) | 9.6 (1.1) | 8.4 (0.9) | 28.2[2] | 10.9 (0.9) | 30.2[2] | 4.9 (0.8) | 16.4 (1.9) | 11.4 (1.9) | 24.7 (1.2) | 8.8 (1.5) |
| Provider asked if child received vitamin A within 6 months | 4.3 (3.4) | 7.2 (0.9) | 2.9 (0.6) | 21.9 (1.8) | 2.2 (0.4) | 30.7 (1.9) | 0.8 (0.2) | 16.7 (1.6) | 32.3 (2.8) | 2.7 (0.4) | 9.3 (1.6) |
| Provider asked if child received any deworming | 0.3 (0.5) | 11.7 (1.2) | 2.9 (0.6) | 20.2 (1.7) | 2.4 (0.4) | – | 3.5 (0.7) | – | 28.3 (2.6) | 4.7 (0.5) | – |
| Provider checked palms/ conjunctiva/mouth for pallor | 32.3 (6.6) | 72.3 (1.7) | 48.1 (1.8) | 57.1 (2.5) | 46.9 (1.5) | 39.1 (2.2) | 14.5 (1.3) | 35.6[2] | 18.1 (2.1) | 39.6 (1.4) | 72 (2.5) |
| Provider asked/caretaker mentioned if child unable to drink or breast feed | 13.7 (5.2) | 23.5 (1.5) | 21.3 (1.4) | 49.4 (2.3) | 28.2 (1.3) | 43.3[2] | 22.9 (1.6) | 25.5 (1.5) | 2.7 (0.8) | 28.5 (1.2) | 59.8 (2.7) |
| Provider asked about normal (breast)feeding pattern | 17.8 (5.7) | 32.6 (1.8) | 38.8 (1.7) | 42.9 (2.3) | 16.8 (1.1) | 43.7 (2.1) | 15.9 (1.4) | 30.3 (1.8) | 26.2 (2.4) | 28.3 (1.2) | 44.6 (2.8) |
| Provider asked about (breast) feeding pattern during this illness | 14.4 (5.6) | 27.5 (1.7) | 30.5 (1.6) | 45.8 (2.2) | 17.4 (1.1) | 45.2 (2.1) | 15.7 (1.3) | 44.8[2] | 6.3 (1.2) | 18.5 (1.0) | 39.6 (2.8) |

Continued

**Table 4** Continued

| Provider competence tracer items | AF | DRC* | HT | KE* | MW | NM* | NP | RW* | SN | TZ | UG* |
|---|---|---|---|---|---|---|---|---|---|---|---|

SEs are in parentheses. AF: Afghanistan 2018, DRC: Democratic Republic of Congo 2018, HT: Haiti 2017, KE: Kenya 2010, MW: Malawi 2013, NM: Namibia 2009, NP: Nepal 2015, RW: Rwanda 2007, SN: Senegal 2017, TZ: Tanzania 2015, UG: Uganda 2007.

Shaded vaues signifies that no data are available.

*Countries with older Service Provision Assessment surveys (prior to 2013 so data may not be comparable). Analysis is limited to primary (non-hospital) health facilities. Competence of providers in Maternal services was only assessed in facilities offering antenatal care services. Competence of providers Child services was assessed in facilities offering services for under-five children.

ANC, antenatal care.

with advice or counselling about growth patterns, developmental milestones or dietary advice which is critically important for mitigating both undernutrition and overweight. The WHO child growth standard also recommends that children's weight and height are plotted on the standard growth chart following a physical examination, as they can be good predictors of dietary status and future disease risk when followed over time.[16]

Explaining side effects of iron by providers was also one of the most suboptimally performed activities from both the patient and the provider perspectives. Iron supplements are required to be taken daily, and empirical research shows that taking iron supplements is often accompanied by side effects such as constipation, cramps, etc, making treatment adherence difficult.[17] It is important that providers are trained to forewarn anaemic pregnant women of both the benefits and side-effects to attain treatment compliance and better health outcomes.[17 18] Failure to counsel women on the proximal and long-lasting maternal, neonatal, and child health benefits of iron supplementation and breast feeding during ANC represents a missed opportunity for improving maternal and child health.[19] It is also important to consider that from a measurement perspective, many nutrition services and campaigns can be based in the community, which might explain the notably better performance in the patients' knowledge and awareness of iron side effects, incongruent to the levels of provider competence in this analysis. Since the SPA data used in the present analysis only focuses on facility-based nutrition services, future research and tools should consider capturing the provision of community-based nutrition services.

This analysis also suggests that patient health literacy and confidence may be a determinant of receiving high-quality nutrition services. Positive patient perception of patient–provider interactions and quality of care is associated with better health experiences.[20] Over 70% of women in 2 out of 11 countries reviewed showed that there was a lack of providers who encouraged women to ask questions during ANC. Positive provider interaction with mothers in the healthcare environment has been shown to be psychodynamic and therapeutic, thereby leading to greater patient satisfaction with the quality of care.[21] To encourage question-asking behaviour, it is important that health providers use simple and empathic language when dealing with expectant mothers and allow them to express themselves without fear.[22]

Our study should be interpreted considering its limitations. First, we included a comprehensive sample of the latest available SPA surveys at the time of the analysis. Included data came from different geographic contexts and time periods, which might have introduced potential bias in the results. Specifically, countries with more recent SPA surveys could have benefited from updated knowledge and the recent advancements in nutrition services compared with countries with older SPA surveys. Our study is also descriptive in nature; primarily aiming

at highlighting the quality of nutrition services at the PHC level in each of the studied countries rather than cross-country comparison. Therefore, we did not use inferential statistics to statistically test or account for the social and/or temporal variations across countries. In addition, we defined PHC as services provided in outpatient consultations in public and private non-hospital SPA facilities, while countries may vary in their definition of what constitute a PHC system. Meanwhile, data was not available for few tracer items in Namibia, Rwanda and Uganda. However, missing data was less than 20% of the total number of tracer items so these three countries were not dropped from the analysis. To minimise potential bias in the present findings, we reported the results for each individual tracer item per country whenever data was available. Future studies may use mixed-effect models that incorporate sociodemographic and temporal factors to determine whether the observed differences between countries were statistically significant.

Comprehensive SPA surveys continue to be essential tools in the assessment of health service delivery in LMICs. However, the surveys lack an array of vital evidence-based nutrition interventions such as calcium, balanced energy, and protein, multiple micronutrient supplementation in pregnancy. Up to our knowledge, there is no validation studies that document the validity and the reliability of the SPA tools/instruments in assessing the quality-of-service delivery of nutrition services at PHC level. Further, facility surveys generally lack specificity in topics key to nutrition assessment and training, especially in non-communicable disease services.

Notably, our findings give rise to additional questions in understanding and framing the drivers influencing the provision of quality nutrition services in PHC. While the availability of nutrition services through investments in system inputs (drugs, diagnostics, equipment, workforce and information systems) is a cornerstone for quality of care, service delivery processes such as having competent providers and clients who are engaged in their care is key to achieving the effective coverage for nutrition services.[23 24] Future research should also consider the use of available SDI to understand the relationship between the quality of nutrition service delivery and nutrition-related health outcomes in LMICs.

## CONCLUSION

Although provider competence and patients' experience across the 11 assessed countries demonstrated the basic delivery of nutrition services, the apparent deficiency in the extent and depth of questions asked for the majority of assessed tracer activities reveal significant opportunities for improving the quality of nutrition service delivery process at the PHC level. Measuring the quality of nutrition services in PHC also remains a challenge due to the limited nature of some of the existing indicators; however, multiple opportunities exist for improving and using the current tools.

**Contributors** MR, TBM and MV-U were responsible for the conception of this study and for methods development. MR and TBM analysed data. MR, TBM, LO and CF prepared the first draft. MR, LO, CF, MV-U all contributed to subsequent iterations of the draft. All authors reviewed results or reviewed and contributed to the final manuscript. MR is responsible for the overall content as a guarantor.

**Funding** This research was supported, in whole or in part, by the Bill and Melinda Gates Foundation (Grant number INV000932), and it had no role in study design, data collection, data analysis, data interpretation or writing of the manuscript.

**Competing interests** None declared.

**Patient and public involvement** Patients and/or the public were not involved in the design, or conduct, or reporting, or dissemination plans of this research.

**Patient consent for publication** Not applicable.

**Ethics approval** Data included in the present analysis were extracted from anonymised publicly available datasets and no patient consent was needed.

**Provenance and peer review** Not commissioned; externally peer reviewed.

**Data availability statement** The original datasets analysed during the current study are available from the DHS programme website, https://dhsprogram.com/data/available-datasets.cfm. The analytical dataset generated during the current study is available from the corresponding author on reasonable request.

**ORCID iDs**
Marwa Ramadan http://orcid.org/0000-0003-4953-7346
Latifat Okara http://orcid.org/0000-0002-3015-6749
Cameron Feil http://orcid.org/0000-0002-9689-7240
Manuela Villar Uribe http://orcid.org/0000-0003-3386-4414

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
