## [Reviewer comments · BMJ Open]

ARTICLE DETAILS

TITLE (PROVISIONAL)	Existing gaps and missed opportunities in delivering quality nutrition services in primary healthcare: a descriptive analysis of patient experience and provider competence in 11 low and middle-income countries
AUTHORS	Ramadan, Marwa; Muthee, Tonny B; Okara, Latifat; Feil, Cameron; Villar-Uribe, Manuela

VERSION 1 – REVIEW

REVIEWER	Ramu Rawat International Institute for Population Sciences
REVIEW RETURNED	20-Aug-2022

GENERAL COMMENTS	The comparison among all countries is justify?, because all countries are belonging in various socio-economic and demographic profile? The people are unable to provide adequate food to their family in many African countries. The food behaviors may effects nutrition conditions of family members. Its required to justify selection of countries in details. Thank you and all the best.
--

REVIEWER	Phillip Phan Johns Hopkins University
REVIEW RETURNED	18-Sep-2022

GENERAL COMMENTS	Thank you for the opportunity to comment on your manuscript. Below are my comments, in no order of importance. I hope they are useful to you. 1. This is a survey report of maternity health in 11 LMICs comparing reports on the patient experience and provide competence. In the chart (please label your charts), 4 of the countries reported that patient experience was lower quality and provider competence, while the reverse was true for the other countries. Why?2. These data came from different time periods. The analysis does not adjust for this, which potentially introduces a bias in the results. One could imagine that those countries reporting more recent data would have benefited from updated knowledge, training programs initiated by the WHO and other health organizations to improve maternal health, and even technology innovations focused on material health. Therefore, your comparisons need to account for the time difference.3. Simple comparisons are insufficient as a standard of reporting. Please provide statistical tests of the differences by country, adjusting for multiple comparisons, of the outcomes of interest.4. You reported two instruments to measure provider competence for child and material health. Please provide the tests of reliability and validity, if appropriate, for these scales. This is a standard
--

	procedure when using and reporting scales. Please do the same for the patients' experience. 5. Please provide references to support the use of your measurement instruments. Have these been used in the past and how have these instruments performed, psychometrically and in comparison with other similar instruments? Thank you again for the opportunity to read and review your work.
--	---

VERSION 1 – AUTHOR RESPONSE

Response to reviewer 1:

Comment	Response /Changes
The comparison among all countries is justify? because all countries are belonging in various socio-economic and demographic profile? The people are unable to provide adequate food to their family in many African countries. The food behaviors may effects nutrition conditions of family members. Its required to justify selection of countries in details. Thank you and all the best.	We agree with the reviewer that the included countries belong to various social and demographic contexts. Our inclusion criteria were based on a comprehensive sample of all low- and Middle-income countries that conducted a SPA survey [for which data was available]. We provided the following sentence in the manuscript “A total of 12 countries completed a SPA survey from 2007 to 2019, 11 of which had sufficient data for the present analysis.” [Under data sources -methods section]. However, we would like to clarify that this study is descriptive in nature; initially aiming at describing the status of provider competence and patient experience in each of the included countries. Scores were calculated and reported for each country separately after applying country-specific survey weights. We did not utilize inferential statistics to make cross-country comparisons or report that observed differences between were statistically significant. In upcoming papers, we may utilize a multi-level modeling approach where we include population and facility level characteristics to account for temporal and geographic variations. To address the reviewer’s comment, we added additional paragraphs/clarifications as recommended by the reviewer in the following section  • Strengths and limitation section” This study is descriptive in nature; inferential statistics were not utilized to account for temporal and geographic variations across countries. [Lines 47-48 in the updated manuscript] • Discussion section [Lines 245-259 in the updated manuscript]

Response to reviewer 2:

Response /Changes	Response /Changes
1. This is a survey report of maternity health in 11 LMICs comparing reports on the patient experience and provide competence. In the chart (please label your charts), 4 of the countries reported that patient experience was lower quality and provider competence, while the reverse was true for the other countries. Why?	We would like to thank the reviewer for highlighting this observation (the difference between subdomain scores). One possible reason is that 3 out of those 4 countries (Namibia, Rwanda, and Uganda) were missing tracer items for provider competence and the subdomain score was calculated as a summary of the remaining items [as indicated in our data analysis section]. We did not drop these 3 countries from the analysis since the number of missing tracer items were less than 20 % of the total number of tracer items. In addition, all four countries had SPA survey prior to 2013 compared to the rest of countries so scores may not be comparable. For clarification, we labelled the figure as recommended by the reviewer. In addition, we added a footnote to figure 1 explaining the difference for these countries and elaborated on it as part of our results [lines 154-158 in the updated manuscript] as well as in the methodological limitations in the discussion section [Lines 2543-259 in the updated manuscript]
2. These data came from different time periods. The analysis does not adjust for this, which potentially introduces a bias in the results. One could imagine that those countries reporting more recent data would have benefited from updated knowledge, training programs initiated by the WHO and other health organizations to improve maternal health, and even technology innovations focused on maternal health. Therefore, your comparisons need to account for the time difference.	We completely agree with the reviewer that more recent data may have better scores and temporal factors may play a role. However, we would like to highlight that we were primary aiming at describing the status of provider competence and patient experience with nutrition services incorporated within PHC in each of the studied countries rather than making cross-country comparisons. We did not utilize inferential statistics, so it was not possible to account for temporal variations. To address the potential bias raised by the reviewer, we added a footnote to our descriptive tables highlighting countries with surveys before 2013, we refrained from reporting any ranking or statistical difference across countries [as indicated by track changes]. In addition, we highlighted such limitation in our discussion section [Lines 246-249 in the updated manuscript]
3. Simple comparisons are insufficient as a standard of reporting. Please provide statistical tests of the differences by country, adjusting for multiple comparisons, of the outcomes of interest.	We would like to thank the reviewer for his insightful comments to improve the quality of this manuscript. We agree with the reviewer regarding the importance of adding statistical tests when making cross-country comparisons. However, the primary aim was not to statistically compare countries given the limited availability of data on adjustment factors and the fact that we did not utilize mixed-effect models for the present

	analysis. However, to address the reviewer recommendation, we modified our table 3 and 4 to include the standard error for each tracer item per country as indicated by track changes.
4. You reported two instruments to measure provider competence for child and maternal health. Please provide the tests of reliability and validity, if appropriate, for these scales. This is a standard procedure when using and reporting scales. Please do the same for the patients' experience.	We would like to clarify that The SPA was designed and further updated in 2022 in collaboration with technical experts, representatives from WHO, UNICEF, Ministries of Health, and others, to make the SPA indicator-driven and refocused on quality of care. The previous versions of the SPA survey (along with its various instruments) do not have publicly available reliability or validity assessment. The scales presented in the current study were a summary score of the key tracer items recommended by literature and previous Demographic Health Survey (DHS) analytic reports as indicated in our data analysis section. To address the reviewer comment, we elaborated more on the use of these scales and their limitations in our discussion section [lines 260-265]
5. Please provide references to support the use of your measurement instruments. Have these been used in the past and how have these instruments performed, psychometrically and in comparison, with other similar instruments?	We added the reference to the DHS website where SPA questionnaires are hosted. The utilized tracer items have also been reported by King, et al as well as utilized by a previous Demographic health survey (DHS) report [references [11,12,15] are cited in the text]. We would like to clarify that patient experience measured here refers to the clients reporting on the services that were offered to them using client exit interview. It does not measure patients' satisfaction or attitude towards the services provided, therefore up to our knowledge, no psychometric analysis was performed for the assessment of nutrition services using these tools in the past.